# *Mycobacterium tuberculosis* Rv2387 Facilitates Mycobacterial Survival by Silencing TLR2/p38/JNK Signaling

**DOI:** 10.3390/pathogens11090981

**Published:** 2022-08-27

**Authors:** Wu Li, Wanyan Deng, Nan Zhang, Huijuan Peng, Yi Xu

**Affiliations:** 1The Joint Center for Infection and Immunity, Institute of Pediatrics, Guangzhou Women and Children’s Medical Center, Guangzhou Medical University, Guangzhou 510623, China; 2Institut Pasteur of Shanghai, Chinese Academy of Sciences, Shanghai 200031, China; 3Key Laboratory of Regional Characteristic Agricultural Resources, College of Life Sciences, Neijiang Normal University, Neijiang 641100, China

**Keywords:** mycobacteria, apoptosis, macrophages, intracellular survival, Rv2387

## Abstract

*Mycobacterium tuberculosis* (Mtb) can evade antimicrobial immunity and persist within macrophages by interfering with multiple host cellular functions through its virulence factors, causing latent tuberculosis. The Rv2387 protein has been identified as a putative effector that potentially participates in Mtb pathogenicity. To explore the role of the Rv2387 protein in host–mycobacteria interactions, we established recombinant *M. smegmatis* strains and RAW264.7 cell lines that stably express the Rv2387 protein. We found that this protein suppresses mycobacteria infection-induced macrophage apoptosis by inactivating caspase-3/-8, thus facilitating the intracellular survival of mycobacteria. In addition, Rv2387 inhibits the production of inflammatory cytokines in macrophages by specifically suppressing TLR2-dependent stimulation of p38 and JNK MAPK pathways. Moreover, we further determined that the Rv2387 protein conferred a growth advantage over recombinant *M. smegmatis* and suppressed the inflammatory response in a mouse infection model. Overall, these data suggested that Rv2387 facilitates mycobacteria to escape host immunity and might be an essential virulence factor in Mtb.

## 1. Introduction

Tuberculosis (TB), an ancient and deadly infectious disease caused by *Mycobacterium tuberculosis* (Mtb), is still a major worldwide health threat [1]. It claims approximately 1.2 million lives per year, and over 10 million new patients were reported in 2020 [2], although intensive chemotherapy has been applied for TB treatment for decades. Due to an increasing prevalence of antibiotic-resistant Mtb, it is important to identify novel strategies for treating Mtb infection, which depends on thorough knowledge of the pathogenesis and immune evasion during Mtb infection [3].

Macrophages are the first defense that confront Mtb. The active apoptosis of macrophages infected with mycobacteria represents an efficient mechanism to eliminate invading intracellular mycobacteria [4,5,6]. During non-pathogenic mycobacteria (such as *Mycobacterium fortuitum* and *Mycobacterium smegmatis*) infection, macrophages undergo rapid and massive apoptosis, which initiates a direct intracellular mycobacteria killing program [7]. In contrast to non-pathogenic mycobacteria, Mtb, a successful and efficient human pathogen, has evolved to express virulence factors to inhibit macrophage apoptosis [8,9,10,11]. These anti-apoptotic virulence factors include SodA, NuoG, Eis, Rv3364c, and Rv3033, exerting their functions by limiting host ROS production, inhibiting the mitochondria pathway, or the TNF-α-mediated pathway [11,12,13,14,15,16]. Thus, thoroughly comprehending the strategies that Mtb deploys to manipulate the apoptotic response to facilitate its survival and infection is fundamental in developing potent anti-TB treatment.

Rv2387 is a conserved hypothetical protein that is selectively present in pathogenic mycobacteria [17]. Our previous study found that Rv2387 is a putative effector involved in the host–Mtb interaction [18]. At the same time, the specific contribution of Rv2387 protein during infection remains unknown. To dissect the role of the Rv2387 protein, recombinant *M. smegmatis* strains and RAW264.7 cells that stably express Rv2387 were constructed. Our data showed that the Rv2387 protein suppresses mycobacteria infection-induced apoptosis and promotes the survival of mycobacteria in primary mouse macrophages, macrophage cell lines, and mice. Additionally, Rv2387 can alter the secretion of cytokines in infected macrophages and mice and inhibit the stimulation of p38 and JNK MAPK pathways. This inhibition effect is dependent on TLR2 and IRAK4. Collectively, these data suggest that the Rv2387 protein interferes with the host immune response against mycobacteria infection by blocking TLR2-mediated inflammatory and anti-apoptotic signaling, representing a potential drug target against Mtb infection.

## 2. Materials and Methods

### 2.1. Bacteria, Cell Lines, and Animals

The *rv2387* gene was amplified by primers 2387-261F (5′-CGG GAA TTC ATG CTG CAT GAG TTC) and 2387-261R (5′-CGT AAG CTT TCA CAG ATC CTC TTC AGA GAT GAG TAT CTG CTC ACC GAT CGA AGC CCC G) from genomic DNA of the *M. tuberculosis* H37Rv strain (Tuberculist database). Then, the *rv2387* gene was ligated to the pMV261 plasmid, an *E. coli*-Mycobacteria shuttle vector [19]. The pMV261 and pMV261-rv2387 plasmids were transformed into *M. smegmatis* mc^2^155 (gifted by Jianping Xie, Southwest University, China). The recombinant Msmg-2387 strains were generated to express an Rv2387-Myc fusion protein stably, while Msmg-EV had the empty vector as a control strain. *M. smegmatis* and its derivative strains were cultured in BD Difco Middlebrook (MB) 7H9 (BD Biosciences, Franklin Lakes, NJ, USA) broth, containing 0.1% (*v*/*v*) Tween 80 (Sigma-Aldrich, St. Louis, MO, USA), 0.2% (*w*/*v*) glucose (Sangon Biotech, Shanghai, China), and 0.2% (*v*/*v*) glycerol (Sangon, China). *Mycobacterium bovis* BCG (gifted by Jianping Xie) was grown in MB 7H9 medium, containing 10% OADC (BD Biosciences, USA), 0.1% (*v*/*v*) Tween 80, and 0.2% glycerol. The *Escherichia coli* (*E. coli*) DH5α strain was cultured in Luria Bertani broth. Kanamycin (Sangon, China) was added to bacterial media as required (25 mg/L for M. smegmatis strains, 50 mg/L for *E. coli* strains).

The human macrophage-like cell line THP-1, obtained from the China Center for Type Culture Collection (CCTCC), was grown in RPMI 1640 culture medium (Gibco, Gland Island, NY, USA), containing 10% fetal bovine serum (FBS, Gibco, USA), 100 U/mL penicillin (Gibco, USA), and 100 mg/mL streptomycin (Gibco, USA). Human embryonic kidney-293 cell lines (HEK-293T) and RAW264.7 macrophage cell lines, also obtained from CCTCC, were cultured in DMEM medium (Gibco, USA), supplemented with 10 mM glutamine (Invitrogen), 10% FBS, 100 mg/mL streptomycin, and 100 U/mL penicillin.

The female 6–8 weeks old C57BL/6 mice were purchased from Vital River Laboratories (Beijing, China) and bred under specific pathogen-free conditions at Institut Pasteur of Shanghai (IPS). All experimental procedures were issued by the Institutional Animal Care and Use Committee of IPS and followed the Guidelines for the Care and Use of Laboratory Animals (Ministry of Health, Beijing, China, 1998).

### 2.2. Construction of Stable Cell Line

The *rv2387* gene was sub-cloned from the pMV261-rv2387 plasmid. The PCR primers (pB-Rv2387F: 5′-GAT ACG CGT ATG CTG CAT GAG TTC TGG GTG and pB-Rv2387R: 5′-TAT ATG CAT CTA ACC GAT CGA AGC CCC GGT) contained MluI and NsiI restriction sites. The purified PCR products were digested by indicated restriction enzymes and ligated to a pHAGE-BSD lentiviral plasmid to construct pHAGE-BSD-Rv2387. The mixes of pHAGE-BSD-Rv2387, envelope vector pMD2.G, and packaging vector psPAX2 in a 5:2:3 ratio were transfected into 293T cells, and the resulting lentivirus was collected. RAW264.7 cells were infected with pHAGE-BSD and pHAGE-BSD-Rv2387 lentiviral particles under 8 mg/L of polybrene and then selected by adding 5 mg/L of blasticidin. RAW264.7 macrophages infected with pHAGE-BSD and pHAGE-BSD-Rv2387 lentiviruses were called RAW-Vec and RAW-Rv2387, respectively.

### 2.3. Isolation of Mouse Murine Bone Marrow-Derived Macrophages (BMDMs)

BMDMs were prepared as described elsewhere [20]. Shortly, bone marrow cells harvested from the tibias and femurs of 6–12 weeks old C57BL/6 mice were incubated in RPMI 1640 medium in the presence of 50 ng/mL M-CSF (Sigma, USA), 10% heat-inactivated FBS, 10 mM glutamine, 1% penicillin, and 1% streptomycin. The media were changed on the third day. All the floating cells were discarded on day seven, and the adherent BMDMs were re-plated the day before the experiment.

### 2.4. Intracellular Survival Assay

RAW264.7 cells, THP-1 cells, and BMDMs were reseeded at a density of 5 × 10^5^ cells in a 12-well plate. THP-1 cells were stimulated with 50 nM PMA (Sigma, USA) for 24 h and cultured for another 24 h with fresh medium. THP-1 macrophages were then infected with recombinant Msmg-2387 or Msmg-EV strains (MOI = 10). For the CFU assay of infected RAW264.7 stable cell lines, RAW-WT, RAW-Vec, and RAW-Rv2387 macrophages were seeded on 12-well plates. The adherent monolayer of macrophages was infected with wild-type *M. smegmatis* (Msmg-WT) or BCG strains. At four hours post-infection, infected macrophages were washed two times and cultured for an additional one hour in RPMI 1640 or DMEM medium, containing 200 mg/L gentamicin, to eradicate extracellular bacilli. At indicated times post-infection, macrophages were washed three times with warmed PBS and lysed in water with 0.1% Triton 100 (Sangon, China). Diluted cell lysates were spread on BD Difco 7H10 plates and cultured for 3–4 days (*M. smegmatis*) or 3 weeks (BCG) in a 37°C incubator. The CFUs were determined by calculating the number of colonies on each petri dish. For measuring the invasion of recombinant *M. smegmatis*, macrophages were infected at an MOI of 25 to allow a more accurate estimate of entry. At four hours post-infection, infected macrophages were washed, lysed, and plated onto 7H10 agar to enumerate the CFUs. The invasion ratio represents the ratio of CFUs count at four hours after infection to the CFU count used for infection. 

### 2.5. Real-Time Quantitative PCR Assay

Murine RAW-Vec or RAW-Rv2387 macrophages were seeded and cultured for 24 h. The macrophages were then infected with Msmg-WT (MOI = 10). Total RNA was extracted by Trizol (Thermo Fisher Scientific, Cleveland, OH, USA) and transcribed into cDNA with oligo(dT18) prime (EZBioscience, Roseville, MN, USA). The real-time qPCR assay was performed by Color SYBR Green qPCR Mix (EZBioscience, Roseville, MN, USA) and ABI QuantStudio 6 FLEX. The primers used here are listed in Table 1. GAPDH was used as an internal control. The relative mRNA expression levels were evaluated by the 2^−ΔΔCT^ method.

### 2.6. ELISA Assay

The protein concentration of cytokines, viz tumor necrosis factor-α (TNF-α), interleukin-10 (IL-10), IL-1β, and IL-6, in the supernatants of infected macrophages or mice serum were estimated using ELISA kits obtained from eBioscience (San Diego, CA, USA), according to the recommended protocol. 

### 2.7. Lactate Dehydrogenase (LDH) Determination

The extent of cell lysis was assessed by measuring the amount of LDH released from the infected macrophages. THP-1 cells were differentiated and infected with engineered Msmg-2387 or Msmg-EV strains (MOI = 10). Culture supernatant was collected at 6, 12, 24, and 48 h post-infection. The relative levels of LDH activity were estimated by a CytoTox96 Nonradioactive Cytotoxicity Assay (Promega Corporation Madison, WA, USA). Percentages of LDH release were calculated, as previously described [21].

### 2.8. Necrosis Assay

To distinguish whether the death of macrophages is apoptosis or necrosis, we infected THP-1 macrophages with Msmg-2387 or Msmg-EV strains (MOI = 10). The culture supernatant was harvested, and the remaining cells were washed and lysed at 6, 48, and 72 h post-infection. The amount of mono- and oligonucleosomes in both whole-cell lysates and culture supernatants were analyzed via a quantitative cell death ELISA kit (Roche Applied Science, IN, USA). The extent of nucleosome enrichment in different fractions was calculated as previously described [22]. Enrichment of nucleosomes in cell lysates was considered a measure of apoptosis, while enrichment in culture supernatants was used as a measure of necrosis.

### 2.9. Flow Cytometry

Apoptotic macrophages were determined by an Annexin V-FITC Apoptosis Detection Kit (Beyotime, Shanghai, China). Briefly, cultured macrophages were harvested 24 h post-infection with recombinant Msmg-2387 or Msmg-EV strains. Macrophages were then washed two times with pre-cold saline and reacted with 200 μL of binding solution, with 5 μL of Annexin V-FITC and 10 μL of propidium iodide (PI). The early apoptotic macrophages were labeled by Annexin V-FITC^+^ and PI^-^; the late apoptotic macrophages were labeled by Annexin V-FITC^+^ and PI^+^ [11]. The apoptosis rates were estimated using a flow cytometer (Beckman CytoFLEX Coulter, CA, USA).

### 2.10. TUNEL Assay

One milliliter of THP-1 cells at 3 × 10^5^ cells/mL density was seeded, differentiated by PMA, and infected with engineered Msmg-2387 or Msmg-EV strains (MOI = 10) for 24 h. TUNEL staining was analyzed via a one-step TUNEL kit (Beyotime, China). In brief, macrophages were washed two times and fixed in a 4% paraformaldehyde solution (Beyotime, China) for 0.5 h. Next, the macrophages were permeabilized with 0.2% (*v*/*v*) Triton X-100 solution for 20 min and stained with the TUNEL working solution for 1 hour. At last, macrophages were stained with DAPI solution for 8 min in the dark. The images were obtained through a fluorescence microscope.

### 2.11. Western Blot

Infected cells were washed two times and lysed in ice-cold RIPA buffer for 0.5 h. An equal amount of cell lysates was separated using SDS-PAGE, followed by transferring them to the PVDF membrane (Millipore, Billerica, MA, USA). The transferred membrane was blocked in 5% non-fat milk (Sangon, Shanghai, China), followed by incubation with the indicated primary antibodies at 4°C overnight. Finally, the membrane was washed three times with TBST and incubated with relevant HRP-conjugated secondary antibodies (Cell Signaling Technology Company, Danvers, MA, USA) at room temperature for one hour and visualized using the BeyoECL Star kit (Beyotime, Shanghai, China). The primary antibodies, viz anti-ERK1/2 (AM076), anti-p38 (AM065), anti-JNK (AJ518), anti-p65 (AN365), and anti-Myd88 (AF2116), were obtained from Beyotime Company (Shanghai, China). The primary antibodies, viz anti-IRAK4 (#4363), anti-phospho-p38 (#4511), anti-phospho-JNK (#4511), anti-phospho-ERK1/2 (#4370), anti-phospho-p65 (#3033), anti-cleaved caspase-3 (#9664), anti-cleaved caspase-9 (#9505), anti-cleaved caspase-8 (#9748), anti-phospho-IRAK4 (#11927), and anti-β-actin (#4970S), were obtained from Cell Signaling Technology Company. ImageJ image software was used to quantify the bands and β-actin as an internal control.

### 2.12. Evaluation of the Virulence of Msmg-2387 in Mice

The recombinant Msmg-2387 or Msmg-EV strains were dispersed in sterile PBS with 0.1% (*v*/*v*) Tween 80 (PBST) and quantified with CFU counting. The 6–8 weeks old female mice (n = 3 mice/group) were injected with 2 × 10^7^ CFUs per animal of recombinant Msmg-EV or Msmg-2387 strains by intraperitoneal administration, as described previously [22]. Blood was harvested from the eyes of mice at 1, 3, 6, and 9 days post-infection (dpi). Serum was isolated by centrifugation at 3000× *g* for 10 min at room temperature. The protein concentration of cytokines was measured by ELISA assay, according to the recommended protocol.

Bacterial loads in mouse organs were counted at 1, 6, and 9 dpi, respectively. After euthanized, various mouse organs, viz spleens, kidneys, livers, and lungs, were harvested and homogenized in 1 mL PBST using aseptic techniques. Homogenized lysates were serial diluted and spread on the 7H10 plate. The bacterial burden was obtained by calculating CFU on the plate after incubation for 3–5 days.

### 2.13. Statistical Analysis

Data in the figures are represented as the means ± SD (standard deviation) of technical triplicate from one representative experiment. Statistical testing was analyzed using a two-tailed Student’s t-test. Significant differences were defined as *p* values (* *p* < 0.05, ** *p* < 0.01, *** *p* < 0.001).

## 3. Results

### 3.1. Rv2387 Enhanced the Intracellular Survival of Recombinant Msmg-2387 in Macrophages

To determine the function of the Rv2387 protein during infection, we analyzed the intracellular survival of engineered Msmg-2387 or Msmg-EV strains in two macrophage cell lines. Expression of the Rv2387 protein resulted in more intracellular *M. smegmatis* living inside THP-1 macrophages and RAW264.7 macrophages (Figure 1A,B). Meanwhile, no difference was observed in the invasiveness between Msmg-EV and Msmg-2387 at four hours post-infection by CFU counting assay (Figure 1C,D) and SYTO-9-based phagocytosis assay [23] (Appendix A), indicating that Rv2387 inhibited clearance of intracellular *M. smegmatis* from infected cells.

To substantiate the role of Rv2387 in enhancing intracellular mycobacteria survival, we constructed RAW264.7 macrophages that stably express Rv2387 (Appendix A) and infected these cells with either wild-type *M. smegmatis* or BCG. The bacterial burdens in macrophages were assayed via enumerating CFU 24 h after infection. Compared with the control groups (wild-type RAW264.7 or RAW264.7 constructed with an empty vector), RAW264.7 cells that expressed Rv2387 harbored significantly higher bacillary burdens (Figure 1E,F). These data suggested that the Rv2387 protein was capable of enhancing the intracellular survival of *M. smegmatis* or BCG in both murine and human macrophage cell lines.

### 3.2. Rv2387 Suppressed Host Inflammatory Response to Mycobacterial Infection

The abovementioned data and our previous studies have shown that Rv2387 does not affect invasiveness or replication of the recombinant Msmg-2387 strain [24]. Moreover, the Rv2387 protein was located in the cell wall fraction of the engineered Msmg-2387 and may have the opportunity to interact with host components (Appendix A) [25]. Therefore, we speculated that Rv2387-mediated resistance to the clearance of mycobacteria from macrophages might be caused by a blunt host immune response [22,26,27]. To investigate whether Rv2387 subverts the host’s innate inflammatory immune response to mycobacterial infection, we detected the mRNA expression of inflammatory-related cytokines in RAW-WT, RAW-Vec, and RAW-Rv2387 macrophages infected with *M. smegmatis* strains. Data from Figure 2A–D showed a much higher mRNA level of anti-inflammatory cytokine IL-10, while a marked lower mRNA expression level of pro-inflammatory cytokines, viz IL-1β, IL-6, and TNF-α, was detected in infected RAW-Rv2387 cells, as compared with the control groups at the indicated time points. Consistently, this is also the same for the secretion levels of cytokines in PMA-differentiated THP-1 macrophages infected with engineered Msmg-2387 or Msmg-EV strains (Figure 2E–H). These data suggested that Rv2387 negatively regulates the mycobacterial infection-triggered host inflammatory response.

### 3.3. Rv2387 Blocked M. Smegmatis-Induced Macrophage Apoptosis

Excessive pro-inflammatory responses lead to cell death and extensive host tissue damage [28]. Because Rv2387 was shown to inhibit inflammatory immune response, this protein may also prevent macrophage death triggered by mycobacterial infection. We infected THP-1 macrophages with engineered Msmg-2387 or Msmg-EV strains to prove this hypothesis. The cytolysis was determined by quantification of the LDH amount released to the culture medium. THP-1 macrophages infected with the engineered Msmg-2387 strain showed diminished cell death, compared with those infected with the Msmg-EV control strain (Figure 3A). Next, we quantified the number of cell death-associated nucleosomes formed inside cells and released into the extracellular space. A relatively lower number of nucleosomes released from the infected macrophages was detected in the culture medium, while the number of intracellular nucleosomes was substantially increased upon mycobacterial infection (Figure 3B,C), indicating that *M. smegmatis* infection induces macrophage apoptosis instead of necrosis. In addition, Rv2387 inhibited the formation of intracellular nucleosomes (Figure 3B). We assumed that Rv2387 inhibits macrophage apoptosis caused by mycobacterial infection. To further confirm the anti-apoptotic effect of the Rv2387 protein, THP-1 macrophages infected with engineered Msmg-2387 or Msmg-EV strains were incubated with FITC-Annexin V/PI solution, followed by flow cytometry assay. Consistently, macrophages infected with Msmg-2387 exhibited significantly decreased total (AnnexinV^+^, 17% vs. 34%), early (AnnexinV^+^ /PI^−^, 3% vs. 2%), as well as late (AnnexinV^+^ /PI^+^, 15% vs. 31%) apoptotic rations compared with those infected with the Msmg-EV strain (Figure 3D,E). Similarly, the TUNEL staining assay showed that less DNA fragmentation was found in macrophages infected with engineered Msmg-2387 strains than in the control strain-infected cells (Figure 3F,G). Thus, Rv2387 blocked *M. smegmatis*-induced macrophage apoptosis, but not necrosis.

### 3.4. Rv2387 Inhibited Apoptosis by Suppressing the Extrinsic Apoptotic Pathway

Previous studies found that both intrinsic apoptosis (mediated by caspase-9) and extrinsic apoptosis (mediated by caspase-8) are involved in cell host death upon mycobacterial infection [11,29]. Both pathways converge upon executioner caspase-3 activation, leading to apoptotic body formation via cleaving a series of target proteins [11,30]. Non-pathogenic and attenuated mycobacteria, including *M. smegmatis*, BCG, and H37Ra, primarily trigger macrophage apoptosis, whereas virulent Mtb strains prevent the apoptotic process from impairing host defense [9]. Indeed, caspase-3 is activated three hours after *M. smegmatis* infection, as indicated by the cleaved caspase-3 (Figure 4A,B). Expression of Rv2387 abolished caspase-3 activation, consistent with its role in inhibiting mycobacteria infection-triggered macrophage apoptosis. To test the effect of caspase-8 or -9 on the stimulation of executioner caspase-3, we assessed the activities of caspase-8 and -9 in mycobacteria-infected THP-1macphages. As expected, *M. smegmatis* infection triggered both caspase-8 and caspase-9 activation. Interestingly, Rv2387 blocked caspase-8 activation, but did not affect the cleavage of caspase-9 (Figure 4A,C,D). The data revealed that both intrinsic and extrinsic apoptosis pathways are activated in *M. smegmatis* infection, and Rv2387 specifically inhibits the activation of both caspase-8/-3.

### 3.5. Rv2387 Inhibited M. Smegmatis-Induced p38 and JNK Activation

NF-κB and MAPK signaling are crucial for immune activation and regulation during mycobacterial infection [31,32,33,34,35]. The inhibition of the Rv2387 protein on the Msmg-2387 strain infection-induced inflammatory response and macrophage apoptosis may be caused by the NF-κB and MAPK activation blockade. To investigate the role of the Rv2387 protein on NF-κB and MAPK signaling, we performed an immunoblot assay of expression and phosphorylation of ERK, p38, JNK, and NF-κB p65 upon either recombinant Msmg-2387 or Msmg-EV strain infections. These molecules were phosphorylated in response to Msmg-EV infection at various time points, of which the phosphorylation of p38 and JNK, but not ERK1/2 or NF-κB p65, was markedly suppressed by the expression of Rv2387 protein (Figure 5). These data indicated that Rv2387 inhibits mycobacterial-induced activation of p38 and JNK pathways in macrophages.

### 3.6. The Effect of Rv2387 on Suppressing p38 and JNK Activation was Dependent on TLR2 Signaling

Mycobacteria infection-triggered macrophage apoptosis and pro-inflammatory cytokine production largely depend on TLR2 and TLR4, which are responsible for sensing components from invading mycobacteria [20,30]. We, therefore, assessed the effect of Rv2387 on TLR2/TLR4 signaling. To avoid the potential effects of compounds on bacteria, we chose RAW264.7 stable cells for this experiment. RAW264.7 cells overexpressed with Rv2387 or not were treated with TLR2 agonist Pam3CSK4 or TLR4 agonist LPS, and subsequently, the activation of TLR-associated signal proteins was analyzed by immunoblotting. As shown in Figure 6A,B, either Pam3CSK4 or LPS treatment induced robust phosphorylation of Erk1/2, p38, and JNK in RAW264.7 macrophages harboring an empty vector. By contrast, activation of p38 and JNK was explicitly blocked in RAW264.7 cells that express Rv2387 in response to Pam3CSK4 (Figure 6A,B), and the expression of cytokines (IL-10, TNF-α, IL-1β, and IL-6) was also manipulated by Rv2387 in the RAW-Rv2387 cells stimulated with the TLR-2 agonist (Appendix A). The results indicated that Rv2387 inhibits TLR2 signaling-induced stimulation of p38 and JNK pathways. Next, we detected the expression and activation of adaptor proteins (IRAK4 and MyD88), which were downstream of TLR2 and upstream of p38 and the JNK signal, in THP-1 macrophages, infected with either recombinant Msmg-2387 or Msmg-EV strains. Figure 6C showed that Rv2387 suppressed the phosphorylation of IRAK4, but did not affect the protein expression levels of both MyD88 and IRAK4. Similar results were obtained from RAW-Rv2387 stable cells treated with Pam3CSK4 (Appendix A). The data indicated that Rv2387 inhibited TLR-induced p38 and JNK activation by blocking the function of IRAK4.

### 3.7. The Role of Rv2387 Protein Was Evaluated in BMDM Cells and Mice

To characterize the role of Rv2387 on primary cells, we first tested mouse BMDMs. We assessed the intracellular survival of engineered Msmg-2387 or Msmg-EV strains in BMDMs by counting the CFU at the indicated time after infection. As shown in Figure 7A, both strains had a rapidly reduced survival rate at the tested time points. However, Msmg-2387 showed much higher CFUs than the Msmg-EV control strains at 24 and 48 h post-infection. The results indicated that Rv2387 enhanced intracellular survival of recombinant Msmg-2387 strains in infected primary cells, in line with the data from macrophage cell lines.

We also evaluated apoptosis rates in mouse BMDMs. BMDMs were infected with engineered Msmg-2387 or Msmg-EV strains for 24 h, stained, and subjected to FACS analysis. The data indicated that when infected with engineered Msmg-2387 or Msmg-EV strains, the numbers of apoptotic macrophages were significantly higher than in the uninfected cells. Meanwhile, Msmg-2387-infected BMDMs had a lower apoptosis level than macrophage-infected Msmg-EV strains (Figure 7B and Appendix A). The results suggested that the Rv2387 protein suppressed apoptosis induced by mycobacterial infection in primary cells.

We next examined the mice model. The recombinant *M. smegmatis* strains at 2 × 10^7^ CFU were injected into C57BL/6 mice by intraperitoneal administration. The initial bacteria burden was evaluated for Msmg-EV or Msmg-2387 in organs at one dpi. For the Msmg-EV infection control, the bacterial burdens in the lungs, kidneys, livers, and spleens rapidly decreased at 6 and 9 dpi (Figure 7C–F). However, in contrast to the Msmg-EV-infected group, there were much higher bacterial burdens in the lungs, livers, and spleens of the Msmg-2387-infected mice at both 6 and 9 dpi (Figure 7C–E). The number of CFUs of bacilli in the kidneys of Msmg-2387-infected mice was much more significant than those in the control group, only at six dpi (Figure 7F). Here, we also investigated the potential effect of Rv2387 in altering the immune response of infected mice. As shown in Figure 7G–J, Rv2387 inhibited the mycobacterial infection-triggered host inflammatory response in mice. However, visual inspection revealed no macroscopic lesions in the lungs, kidneys, livers, or spleens of the recombinant *M. smegmatis*-infected mice (data not shown). Thus, the Rv2387 protein promoted mycobacteria survival and dampened inflammatory response in mice.

## 4. Discussion

Mtb resides and proliferates primarily in alveolar macrophages of infected humans. The persistence of Mtb within macrophages is realized via several immune evasion strategies, including, but not limited to, suppressing apoptosis, blocking phagosome acidification, and modulating inflammatory responses [10,36,37]. Mtb secretes various virulence factors to mediate its immune escape. Knowledge of how the Mtb-immune system interacts can be translated into TB treatment and TB epidemic control [3]. The Rv2387 protein was predicted as a conserved membrane permease, but its role during Mtb infection remains unknown [38]. It is found that Rv2387 orthologs exist in the genome of pathogenic mycobacteria, while they are absent in non-pathogenic mycobacteria, implying that Rv2387 may play a crucial role in mycobacterial pathogenicity. In this study, we found that Rv2387 functions as a potent immune response inhibitor to facilitate the survival of intracellular mycobacteria.

Interestingly, we previously found that Rv2387 confers susceptibility of recombinant *M. smegmatis* to acidic stress, the main bactericidal agent required to eradicate mycobacteria in macrophages [24]. We speculate that Rv2387 may have different roles in bacterial physiology and host–Mtb interactions, as is the case for the outer membrane channel CpnT, which is a dual functional protein [39]. The well-studied CpnT was reported to be composed of a C-terminal domain (CTD) with glycohydrolase activity and an N-terminal domain (NTD) with channel-forming activity [39,40]. The CTD is required for the survival of Mtb in macrophages [40]. The oligomeric NTD is required for efficient nutrient uptake by Mtb and *M. bovis* BCG [39]. However, CpnT makes the slow-growing mycobacteria more susceptible than its mutant to various TB drugs and to nitric oxide, which is a major bactericidal agent that is required to kill Mtb in vivo [41]. By considering the fact that recombinant *M. smegmatis* harboring *rv2387* is susceptible to acidic stress, we hypothesize that Rv2387 may be involved in the inhibition of phagosome maturation. IL-10 was reported to block phagolysosomal fusion in Mtb-infected macrophages, decreasing pro-inflammatory cytokine production [42,43]. Consistently, we identified that Rv2387 up-regulates the production of IL-10 and down-regulates the expression of pro-inflammatory cytokines. However, the details for this need further investigation. TLR2 and TLR4 have been found responsible for host recognition of invading mycobacteria, stimulating downstream NF-κB and MAPK-mediated inflammatory pathways to establish a host immune response to defense against mycobacterial infection [10,44]. Rv2387 specifically dampens TLR2-dependent p38 and JNK activation upon mycobacterial infection by inhibiting IRAK4 phosphorylation. Whether this inhibition is dependent on the permease activity of Rv2387 and how Rv2387 executes this function require further exploration. Because p38 and JNK signaling pathways are blocked, pro-inflammatory cytokine expression induced by these pathways is markedly downregulated in the presence of Rv2387 in macrophages infected by mycobacteria. It is known that pro-inflammatory cytokines are involved in establishing immune defense and the restriction of Mtb infection [45,46]. The expression of Rv2387, thus, confers the capacity for immune evasion during mycobacterial infection. In addition to the induction of inflammatory response, the MAPK pathway has been reported to trigger ROS production, which acts against the invading Mtb [47,48]. It will be intriguing to investigate whether Rv2387 interferes with the ROS defense system.

Apoptosis is another efficient effector mechanism of the innate immune response against mycobacteria [49,50], as induction of macrophage apoptosis upon infection causes the direct killing of intracellular mycobacteria [51,52]. Non-pathogenic mycobacteria (*M. smegmatis*) or facultative-pathogenic mycobacteria (such as *M. bovis* BCG or *M. kansasii*) induce rapid host cell apoptosis, whereas virulent Mtb strains evade the host cell apoptosis program [7,53]. Several Mtb effectors inhibit caspase-9-mediated intrinsic apoptosis, such as Rv3033 [11], PtpA [54], GroEL2 [55], and MPT64 [56]. Mtb has also adopted a series of effectors to inhibit caspase-8-mediated extrinsic apoptosis, viz NuoG [14], Rv3654c [57], and EspR [20]. We found that Rv2387 inactivates both caspase-8 and caspase-3, thus shutting down the host apoptosis program. The blockade of both initial and effector caspases ensures the inhibitory effect of apoptosis during virulent mycobacterial infection. Thus, Rv2387 plays an essential role in mycobacterial virulence through dampening macrophage apoptosis. Nevertheless, the detailed role of the Rv2387 protein in pathogenicity should be further investigated by the deletion of the gene in Mtb.

In summary, we demonstrated here that the Rv2387 protein can promote mycobacterial intracellular survival in human macrophages, murine macrophages, and mouse models. This effect relies on TLR2 and its downstream pathways. Rv2387 inhibited TLR2 activation, suppressed p38 and JNK signal pathways, and dampened pro-inflammatory responses and macrophage apoptosis. Our work demonstrated that Rv2387 sabotages host immune responses to facilitate mycobacterial infection and could be regarded as a virulence marker of mycobacteria, deepening our understanding of host–mycobacterial interactions and potentially could be translated into the treatment of virulent mycobacterial infections.

## Figures and Tables

**Figure 1 pathogens-11-00981-f001:**
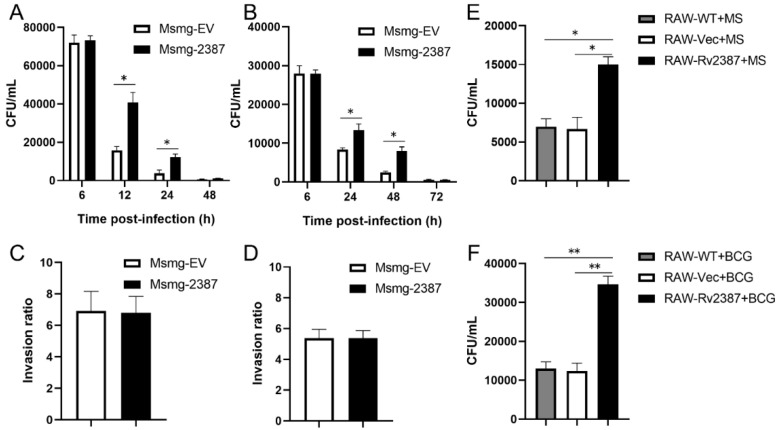
Rv2387 promoted intracellular survival of Mycobacteria. THP-1 macrophages and RAW267.4 macrophages were infected with engineered Msmg-2387 or Msmg-EV strains at an MOI of 10, respectively. The intracellular survival of Msmg-2387 and Msmg-EV strains in THP-1 macrophages (**A**) and RAW267.4 macrophages (**B**) was calculated by CFU counting. The invasion ratio of bacilli for THP-1 macrophages (**C**) and RAW267.4 macrophages (**D**) at an MOI of 25:1 at four hours post-infection was calculated by CFU counting. The numbers of viable intracellular mycobacteria in RAW-WT, RAW-Vec, and RAW-Rv2387 cells infected with wild-type *M. smegmatis* (MS) (MOI = 10) for 24 h (**E**) or BCG (MOI = 10) for 48 h (**F**). Data are means ± SD of technical triplicate from one representative out of three or more independent experiments (* *p* < 0.05, ** *p* < 0.01).

**Figure 2 pathogens-11-00981-f002:**
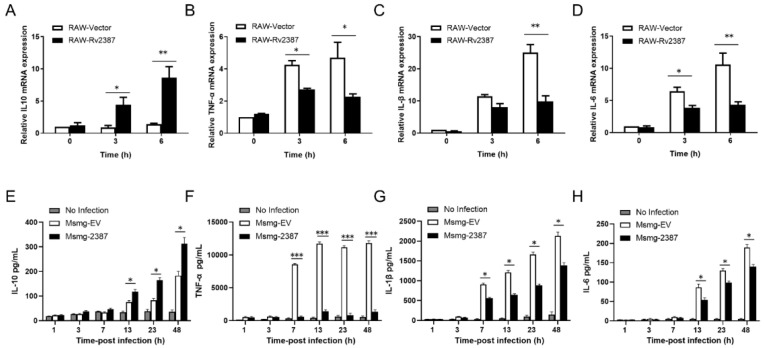
Rv2387 induced IL-10 production, but suppressed TNF-α, IL-1β, and IL-6 production. Murine recombinant RAW-Vec and RAW-Rv2387 macrophages were infected with Msmg-WT strain (MOI = 10) and subject to RNA isolation at the indicated times. The mRNA expression of IL-10 (**A**), TNF-α (**B**), IL-1β (**C**), and IL-6 (**D**) was then analyzed by real-time quantitative PCR assay. THP-1 macrophages were infected with recombinant Msmg-2387 and Msmg-EV strains, respectively. Culture supernatants were collected at different time points and subjected to an ELISA analysis to monitor the secretion of IL-10 (**E**), TNF-α (**F**)**,** IL-1β (**G**), and IL-6 (**H**). Data are means ± SD of technical triplicate from one representative out of three or more independent experiments (* *p* < 0.05, ** *p* < 0.01, *** *p* < 0.001).

**Figure 3 pathogens-11-00981-f003:**
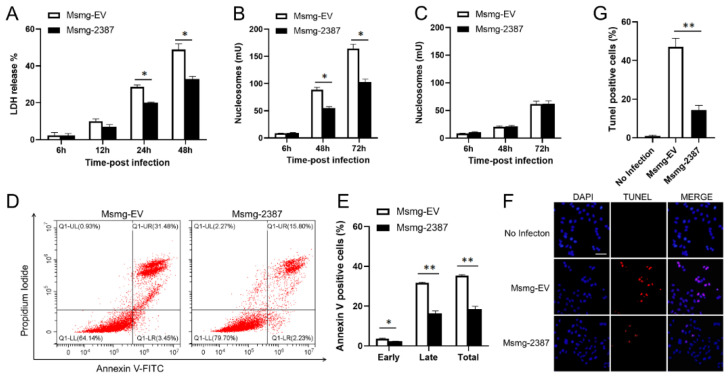
Rv2387 decreased mycobacteria-induced macrophage apoptosis, but not necrosis. Culture supernatants were harvested from THP-1 macrophages infected with engineered Msmg-2387 or Msmg-EV (MOI = 10) at different time points. (**A**) As a measure of cell lysis, the levels of LDH activity in supernatants were detected by a commercial kit. Quantitation of apoptosis (**B**) and necrosis (**C**), as measured by the delivery of nucleosomes to cell culture supernatant or not by a quantitative ELISA assay. (**D**) THP-1 macrophages were infected with either engineered Msmg-2387 or Msmg-EV strains for 24 h. The apoptotic macrophages were determined by flow cytometer stained with FITC-Annexin V/PI. (**E**) Percentages of different types of apoptotic cells (early, late, and total) are enumerated in (**D**). (**F**) DNA fragmentation in infected macrophages was detected via TUNEL staining analysis. The scale bars are equivalent to 50 µm. (**G**) The percentages of apoptotic cells are enumerated in (**F**). Data are means ± SD of technical triplicate from one representative out of three or more independent experiments (* *p* < 0.05, ** *p* < 0.01).

**Figure 4 pathogens-11-00981-f004:**
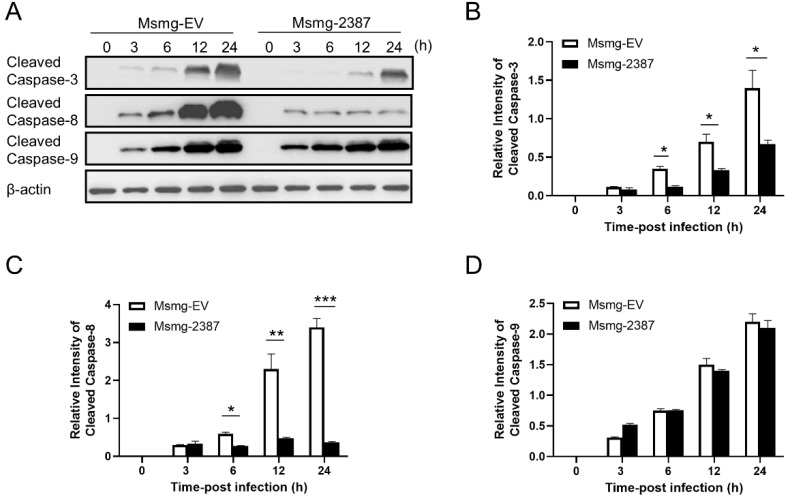
Rv2387 inhibited apoptosis by suppressing the extrinsic apoptosis instead of the intrinsic apoptosis. (**A**) THP-1 macrophages were infected with engineered Msmg-2387 or Msmg-EV (MOI = 10) at different time points. The activation levels of caspase-3, caspase-9, and caspase-8 (cleaved bands) were detected by immunoblotting. (**B**–**D**) Semi-quantification of cleaved bands of caspase-3, caspase-8, and caspase-9. Bands were quantified by ImageJ image software, using β-actin as internal controls. The relative intensity is defined as the ratio of cleaved caspase to β-actin. Similar results were obtained in three or more independent experiments (* *p* < 0.05, ** *p* < 0.01, *** *p* < 0.001).

**Figure 5 pathogens-11-00981-f005:**
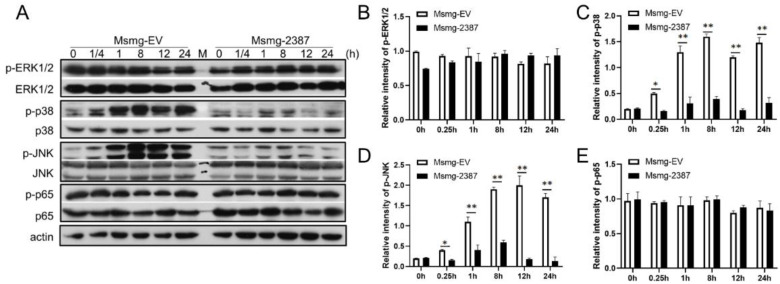
Rv2387 blocked the activation levels of p38 and JNK. (**A**) THP-1 macrophages were infected with engineered Msmg-2387 or Msmg-EV strains (MOI = 10) at different time points. Infected cells were lysed in RIPA buffer and subjected to immunoblot using the antibodies labeled in the figure. (**B**–**E**) Semi-quantification of phosphorylation of the ERK1/2, p38, JNK, and p65 proteins. Bands were quantified by ImageJ image software, using β-actin as internal controls. The relative intensity is defined as the ratio of phosphorylated protein to β-actin. Similar results were obtained in three or more independent experiments (* *p* < 0.05, ** *p* < 0.01).

**Figure 6 pathogens-11-00981-f006:**
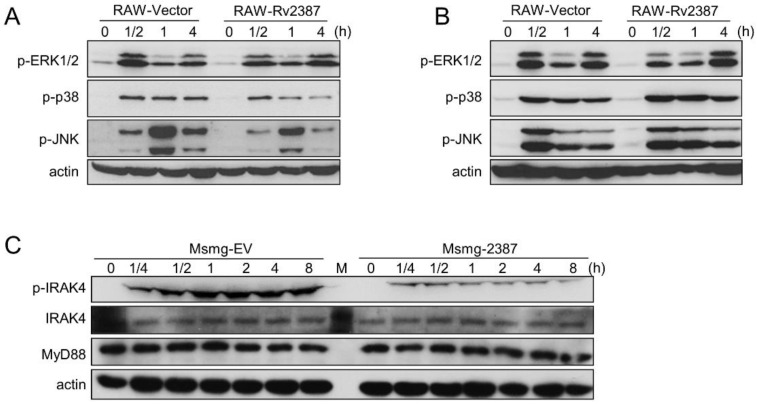
Rv2387 blocked the activation of the p38 and JNK pathway via TLR2 signaling. Murine RAW-Vec and RAW-Rv2387 macrophages were treated with TLR2 agonist Pam3CSK4 ((**A**), 200 ng/mL) and TLR4 agonist LPS ((**B**), 100 ng/mL) at different time points. (**C**) THP-1 macrophages were infected with engineered Msmg-2387 or Msmg-EV strains (MOI = 10) for the indicated time. Infected macrophages were lysed in RIPA buffer and then subjected to immunoblot analysis to detect the protein expression of phosphorylated or total Erk1/2, JNK, p38, MyD88, and IRAK4. Similar results were obtained in two or more independent experiments.

**Figure 7 pathogens-11-00981-f007:**
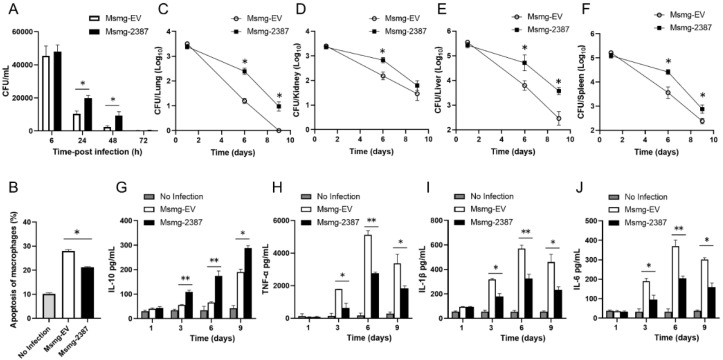
The role of Rv2387 was evaluated in BMDM cells and mice. (**A**) Intracellular survival of engineered Msmg-2387 or Msmg-EV strains after infection of murine BMDMs (MOI = 10). Aliquots of infected macrophages were washed twice and lysed at the indicated time. Cell lysates were then diluted and spread on 7H10 agar. (**B**) Percentages of apoptotic BMDMs (annexin V^+^) infected with engineered Msmg-2387 or Msmg-EV strains (MOI = 10) for 24 h, followed by FACS analysis. (**A**,**B**) Data are means ± SD of technical triplicate from one representative experiment out of three independent experiments. The 6-8 week old C57BL/6 female mice were injected with Msmg-2387 or Msmg-EV using 2 × 10^7^ CFUs per animal by intraperitoneal administration (3 mice/group/time point). Mice were sacrificed at 1, 6, or 9 dpi. Organs were harvested and homogenized in PBST using aseptic techniques. The bacterial burden in lungs (**C**), kidneys (**D**), livers (**E**), and spleens (**F**) of mice was investigated by counting CFUs. At 1, 3, 6, and 9 dpi, serum was collected from mice. The levels of IL-10, TNF-α, IL-1β, and IL-6 (**G**–**J**) in serum were determined via ELISA assay. (**C**–**J**) Data are means ± SD of technical triplicate from one representative experiment out of two independent experiments (* *p* < 0.05, ** *p* < 0.01).

**Table 1 pathogens-11-00981-t001:** Primers for real-time qPCR.

Primers	Sequence
IL-10-Forw	ATGCCTGGCTCAGAC
IL-10-Rev	GTCCTGCATTAAGGAGTCG
IL-1β-Forw	GCAACTGTTCCTGAACTCAACT
IL-1β-Rev	ATCTTTTGGGGTCCGTCAACT
TNF-α-Forw	CCCTCACACTCAGATCATCTTCT
TNF-α-Rev	GCTACGACGTGGGCTACAG
IL-6-Forw	GAGAGGAGACTTCACAGAGGATAC
IL-6-Rev	GTACTCCAGAAGACCAGAGG
Gapdh-Forw	GAAGGGCTCATGACCACAGT
Gapdh-Rev	GGATGCAGGGATGATGTTCT

## Data Availability

The original contributions presented in the study are included in the article/Appendix A. Further inquiries can be directed to the corresponding author/s.

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
