# Peer review of "Mycobacterium tuberculosis Rv2387 Facilitates Mycobacterial Survival by Silencing TLR2/p38/JNK Signaling"

_pathogens, 2022, doi:10.3390/pathogens11090981_

Round 1

Reviewer 1 Report

In this work, the authors have identified the role of a novel protein Rv2387 in Mycobacteria’s defense against the host: survival in the macrophages. The authors have used tested it across macrophage cell lines and in the murine model of infection. The authors have also shown that Rv2387 decreased mycobacteria-induced macrophage apoptosis but there was no effect on necrosis. The authors dissected that the expression of Rv2387 by the pathogen can lead to the failure of Caspase 3-activation and suppression of the extrinsic pathway of apoptosis (mediated by Caspae8). Rv2387 also blocked the activation of p38 and JNK.  Overall, the work is solid and should be of interest to people studying Mycobacteria infection. However, I‘ve couple of questions, the authors should address to make the paper more enriched:

Major Comment

1.     The authors have used an overexpressing strain to dissect the role of Rv2387. However, the authors have not conducted any experiment where Rv2387 was deleted in the genome and see its effects on Mycobacterium tuberculosis survival inside the host by inactivation of apoptotic pathways. The authors should knock out the gene encoding Rv2387 and test it in a murine model of infection as overexpression of a gene may not represent the actual protein titers and lead to ectopic effects that are merely artifacts of the experiments.

2.     The authors have mentioned that Rv2387 leads to the inactivation of Caspase 3 and the extrinsic pathway of apoptosis mediated by Caspase-8. However, the inactivation of Caspase-3 should also influence Caspase-9. However, the authors didn’t detect any change in Caspase-9. The authors should discuss a little bit more on this.

3.     Is Rv2387 an essential gene? In that case, the authors should try to use conditional mutation or knock-down experiments.

Minor comment

In the introduction section line 40 please change the word “secret” to “expresses”.

Round 2

Reviewer 1 Report

The authors have addressed all the queries and made significantly improved the manuscript. They did all the control experiments.

Author Response

We appreciate the Reviewer for his/her effort to review our manuscript, and his/her positive feedback.
